# Metabolic responses of wheat seedlings to osmotic stress induced by various osmolytes under iso-osmotic conditions

Eva Darko [1]*, Balázs Végh[1], Radwan Khalil[2], Tihana Marček[3], Gabriella Szalai[1], Magda Pál[1], Tibor Janda[1]

1 Department of Plant Physiology, Agricultural Institute, Centre for Agricultural Research, Hungarian Academy of Sciences, Martonvásár, Hungary, 2 Botany Department, Faculty of Science, Benha University, Benha, Egypt, 3 Department of Food and Nutrition Research, Faculty of Food Technology, Josip Juraj Strossmayer University of Osijek, Osijek, Croatia

* darko.eva@agrar.mta.hu

**Data Availability Statement:** All relevant data are within the manuscript and its Supporting Information files.

## Abstract

Many environmental stresses cause osmotic stress which induces several metabolic changes in plants. These changes often vary depending on the genotype, type and intensity of stress or the environmental conditions. In the current experiments, metabolic responses of wheat to osmotic stress induced by different kinds of osmolytes were studied under iso-osmotic stress conditions. A single wheat genotypes was treated with PEG-6000, mannitol, sorbitol or NaCl at such concentrations which reduce the osmotic potential of the culture media to the same level (-0.8MPa). The metabolic changes, including the accumulation of proline, glycine betaine (GB) and sugar metabolites (glucose, fructose, galactose, maltose and sucrose) were studied both in the leaves and roots together with monitoring the plant growth, changes in the photosynthetic activity and chlorophyll content of the leaves. In addition, the polyamine metabolism was also investigated. Although all osmolytes inhibited growth similarly, they induced different physiological and metabolic responses: the $CO_2$ assimilation capacity, RWC content and the osmotic potential ($\psi\pi$) of the leaves decreased intensively, especially after mannitol and sorbitol treatments, followed by NaCl treatment, while PEG caused only a slight modification in these parameters. In the roots, the most pronounced decrease of $\psi\pi$ was found after salt-treatments, followed by PEG treatment. Osmotic stress induced the accumulation of proline, glycine betaine and soluble sugars, such as fructose, glucose, sucrose and galactose in both the root and leaf sap. Specific metabolic response of roots and leaves under PEG included accumulation of glucose, fructose and GB (in the roots); sucrose, galactose and proline synthesis were dominant under NaCl stress while exposure to mannitol and sorbitol triggered polyamine metabolism and overproduction of maltose. The amount of those metabolites was time-dependent in the manner that longer exposure to iso-osmotic stress conditions stimulated the sugar metabolic routes. Our results showed that the various osmolytes activated different metabolic processes even under iso-osmotic stress conditions and these changes also differed in the leaves and roots.

**Funding:** This work was funded by the National Research, Development and Innovation Office K112226. The funders had no role in study design, data collection and analysis, decision to publish, or preparation of the manuscript.

**Competing interests:** The authors have declared that no competing interests exist.

## Introduction

Plants are often exposed to different kinds of abiotic stress conditions, like drought or salinity stress that results in diminished plant growth and crop productivity. Most of these environmental stresses impose osmotic stress on plants by reducing the water potential of the environment. The consequences of osmotic stress manifest in the inhibition of cell elongation, stomata closure, reduction of photosynthetic activity, disturbances in water and ion uptake, translocation of assimilates, and changes of various metabolic processes. Wide-ranging studies on the underlying physiological and biochemical mechanisms related to plant responses to osmotic stress have identified several primary and secondary metabolites that are involved in plant stress responses and tolerance (see review in Wang et al.[1]). Osmotic stress triggers the accumulation of amino acids and their derivatives (especially proline and glycine betaine), soluble sugars (such as fructose, glucose, sucrose, galactose, trehalose), sugar alcohols or polyols (e.g. mannitol, sorbitol, inositols) and polyamines (PA), such as putrescine (PUT), spermidine (SPD) and spermine (SPN). However, the synthesis and accumulation of these osmoprotectants strongly depend on environmental conditions, type, strength and duration of stress, and sensitivity of the plants. For instance, the accumulation of proline, GB, γ-aminobutyric acid (GABA), soluble sugars, intermediates of tricarboxylic (TCA) cycle and myo-inositol were found in several wheat genotypes exposed to drought stress, but the amounts of these metabolites depended on the genotype [2,3]. When Do et al. [4] compared the metabolic responses of 21 rice cultivars to salt stress, the PUT levels increased or did not change in tolerant genotypes but decreased in susceptible ones, while SPD showed minor changes and SPN increased in all the genotypes. However, when these plants were exposed to long-term, mild drought stress [5], the amounts of PUT and SPD decreased while SPN increased together with proline and GABA, but there was no correlation between the levels of these metabolites and the drought tolerance of the cultivars. The accumulation of osmoprotective compounds may also be tissue-specific [6]. For instance, when maize was exposed to salt stress, a clear dose-dependent effect was observed in both roots and shoots, which was correlated with the accumulation of sucrose and alanine in both tissues, but there were differences between root and shoot in the accumulation of other osmoprotectants: elevated levels of GB, glutamate and aspartate were found in the shoots together with the lower amounts of malic acid and glucose, while the GABA, malic acid and succinate levels increased in the roots [6]. Since the metabolic changes are affected by many factors including the sensitivity of plants, growth conditions and the type and severity of stress applied, it is really difficult to understand the metabolic changes independently of genotypic and environmental factors.

Application of osmotically active compounds in hydroponic system is a simplified method for studying the effects of osmotic stress. Polyethylene-glycol (PEG), mannitol or sorbitol are widely applied to stimulate osmotic stress [7,8]. NaCl is also added to the nutritional solution when the salt stress response of plants is studied [9]. Usually dose-response curves of certain osmotic agents are analysed, but limited information is available on metabolic changes under iso-osmotic stress conditions. Comparative studies are scarce. Slama et al. [8] compared the effects of mannitol and PEG on growth, water content and solute (proline and total soluble sugars) accumulation in a halophyte plant and found that mannitol caused less injury than PEG. During the treatments, the soluble sugar content remained unchanged while proline increased strongly, especially in mannitol-treated plants [8]. Later, the effects of PEG and NaCl were also compared under iso-osmotic conditions [10,11]. When Ghuge et al. [12] studied the effects of mild (-0.4 MPa) osmotic stress induced by NaCl, PEG or mannitol treatments in potato, the rate of sugar accumulation was similar in all the treatments while salt stress induced

higher proline accumulation than PEG or mannitol and the GB accumulation was higher in the PEG-treated leaves.

The aim of the present study was to identify the common and specific metabolic responses to osmotic stress induced by various osmolytes including NaCl, PEG-6000, mannitol or sorbitol in young wheat plants under iso-osmotic stress conditions. Changes in growth, photosynthetic activity and chlorophyll content were monitored together with proline and GB accumulation and changes in the sugar and polyamine metabolites in both roots and shoots. This approach could contribute to a better understanding of the differences in metabolic responses independently of genotypic and environmental factors.

## Materials and methods

### Plant material and growth conditions

A modern wheat (*Triticum aestivum* L.) cultivar, Mv Béres, developed at the Agricultural Institute, Centre for Agricultural Research, Hungarian Academy of Sciences (Martonvásár), was used in the experiments. Wheat seeds were germinated on water-wetted filter paper for 3 days. Seedlings with similar root length w ere then grown in pots (12 plants/0.6 L) containing modified ½-strength Hoagland solution [13] in a phytotron growth chamber (PGR 15, Conviron, Controlled Environments Ltd., Winnipeg, MB, Canada) under a 16/8h day/night photoperiod at 150µmol $m^{-2}$ $s^{-1}$ PPFD with 22/20˚C day/night temperature for 9 days. After this, control plants continued to grow in modified ½-strength Hoagland solution, which had an osmotic potential of -0.035MPa, while in the stress treatments osmotic stress was induced by adding NaCl (150mM), polyethylene glycol (PEG-6000, 20%), mannitol (300 mM) and sorbitol (300mM) resulting in a reduction of the osmotic potential to -0.8 MPa, similarly as found by Lokhande et al. [14] and Patade et al. [10]. The solutions were renewed every two days and the stress treatments were continued for 6 days. Leaf and root samples were collected every second day before changing the culture medium.

### Determination of plant growth and leaf RWC

The effect of the osmolytes was monitored by measuring the length and weight of roots and shoots of 20 plants collected from different pots in each treatment every two days. The relative water content (RWC) of the leaves was determined by measuring the fresh weight (FW), turgid weight (TW) and dry weight (DW) of leaf segments. Approximately 0.3 g leaves were sliced into 2cm segments for each sample, and 5 samples were prepared for each day and treatment. TW was determined after floating the leaf segments on distilled water for 5 h and DW was measured after drying the leaf segments in an oven at 80 ℃ for 24 h. RWC was calculated according to the equation: RWC (%) = [FW-DW]/(TW-DW] $^*$ 100 with five repetitions per day and treatment.

### Determination of the photosynthetic activity and chlorophyll content of the leaves

The photosynthetic activity and chlorophyll content of the leaves were determined to monitor the physiological responses induced by osmotic stress. A Ciras 2 portable photosynthesis system with a narrow (1.7 $cm^2$) leaf cuvette (Amesbury, USA) was used to measure the net photosynthetic rate (Pn), stomatal conductance (gs) and transpiration rate (E). These parameters were determined at the steady-state level of photosynthesis using a $CO_2$ level of 400µL $L^{-1}$ and light intensity of 550µmol $m^{-2}$ $s^{-1}$. Five randomly selected mature leaves (of similar age) were measured each day for each treatment.

The chlorophyll content of attached leaves was determined with a SPAD-502 chlorophyll meter (Spectrum Technologies, Plainfield, IL, USA). Fifteen fully developed leaves were measured per day and treatment.

## Determination of the osmotic potential of leaf and root sap

Leaf and root sap was isolated to analyse the osmotic potential of the tissues and to determine the concentration of certain metabolites. One gram of frozen leaf and root sample was crushed in liquid nitrogen and the cold powders were centrifuged at 13000 g for 15 min at 4°C in centrifuge tubes containing micro-SpinFilter (Micro-SpinFilter Tubes, Fisher Scientific; 0.45 μm). Aliquots of the tissue saps were stored at -80°C until required. Five isolation processes were performed per day and treatment ensuring five biological replicates for each treatment. The osmotic potential of the leaf and root saps was determined using of an Osmomat 030 freezing-point osmometer (Gonotech, GmbH, Berlin, Germany) and the osmotic potential (ψπ) values were calculated according to Bajji et al. [15].

## Determination of metabolites from leaf and root sap

For metabolic analyses, the root and leaf saps were diluted with ultrapure water. The amount of proline and soluble quaternary ammonium compounds (QAC), including glycine-betaine (GB), were determined from the diluted (1:17.5) supernatants of root and leaf saps (n = 5 for each day and treatment) using a UV-Visible spectrophotometer (160A, Shimadzu Corp, Kyoto, Japan). The determination of proline was based on its reaction with ninhydrin [16] while the QAC content was measured using the periodide method [17]. Briefly, for the determination of proline content, reaction mixtures containing diluted tissue sap, glacial acetic acid and ninhydrin reagents (1:1:1) were placed in a water bath at 100°C for 1h, then cooled on ice. The purplish chromophores were extracted with toluene and the absorbance of the samples was measured at 520 nm. For the determination of soluble QAC, the diluted samples were combined with ice-cold 2N $H_2SO_4$ (1:1) and incubated on ice for 1 h. Then cold KI-$I_2$ reagent was added to the solution and the samples were stored on ice in the dark for 24 h. After centrifugation at 10000 rpm for 10 minutes at 0°C, the KI-$I_2$ solution was removed and the periodite crystals were washed gently with cold distilled water and then dissolved in 1,2-dichloroethan. The absorbance of the samples was read at 365 nm. Proline and glycine-betaine standards purchased from Sigma-Aldrich (Darmstadt, Germany) were used for the quantification.

The sugar content of the tissue saps (n = 5 for each day and treatment) was analysed by HPLC according to Gondor et al. [18]. Briefly, the diluted (1:5) samples were incubated at 96°C for 10 min, then centrifuged at 10000g for 10 min. The sugars were separated on a Supelcosil LC-NH2 column (250 × 4.6 mm, 5 μm, Supelco, Bellefonte, USA) under isocratic elution with 75% acetonitrile, and the sugar components were detected with a differential refractometer (W410, Waters, USA). Standards purchased from Sigma-Aldrich (Darmstadt, Germany) were used for the quantification.

## Determination of polyamine content

The amounts of the most abundant polyamines (PAs), putrescine (PUT), spermidine (SPD) and spermine (SPN) were determined from leaf and root samples collected at the end of the experiment in three repetitions. A modified protocol of Smith and Davies [19] was used for the analyses. Briefly, 0.2 g samples were homogenized with 1mL of 0.2 M perchloric acid, stored on ice for 20 min then centrifuged at 10 000g for 10 min. PAs were determined from the supernatants (named as free polyamines) after producing their dansylated derivatives. These products were separated by HPLC using a W2690 separation module on a reverse phase

column (Kinetex C18, 5μ, 100 × 4.6 mm, Phenomenex, Inc.) and detected with a W474 scanning fluorescence detector (Waters, Milford, MA, USA) with excitation at 340 nm and emission at 515 nm. Standards purchased from Sigma-Aldrich (Darmstadt, Germany) were used for the quantification and the amount of PAs was calculated as μg/g FW.

## Statistics

The results were obtained from three independent experiments with 6 pots of treatment. For the determination of morphological and biochemical parameters, plants or plant tissues were collected from each pots randomly. The values presented in the figures and tables are the mean ± standard deviation. Comparisons between all treatments and the control were made using Tukey's *post hoc* test with the SPSS 16.0 statistical program. Different letters indicate significant differences at the $P < 0.05$ level. Principal Component Analysis (PCA) was performed using the STATISTICA software package (version 13.4) to determine the metabolic responses to different kinds of treatments. PCA was done using records of ten variables. The loadings showed correlations between different principal components (PC) and variables whereby high loadings represented strong correlation (>0.75). PCA plots were plotted separately for the root and shoot.

# Results

## Growth modifications induced by osmotic stress

The growth potential of young wheat plants immediately declined in hydroponic medium with highly negative (-0.8 MPa) osmotic potential. Thereafter, neither the root nor the shoot lengths showed any significant change during the 6 days of treatment, irrespective of the type of osmotically active compound (PEG, NaCl, mannitol or sorbitol) (Fig 1). Similar results were found for plant weight (Fig 2). Mannitol and sorbitol treatments completely inhibited an increase in fresh weight in both roots and shoots while PEG and NaCl treatment resulted in a slight increase in the shoot weight but not in the root weight. These results showed that in the applied concentration, all the osmolytes inhibited the growth rapidly and to a similar extent. However, a detailed analysis of the physiological and biochemical mechanisms revealed several differences.

## Changes in photosynthetic activity and chlorophyll content in the leaves

All the osmolytes caused reduction in the photosynthetic activity of the leaves, which was manifested as a decrease in $CO_2$ assimilation rate (Pn), stomatal conductance (gs, indicating stomatal closure) and transpiration rate (E) (Table 1). The most intense decrease in Pn was detected in the leaves treated with mannitol or sorbitol, followed by the salt treatment. The least reduction was induced by PEG treatment. The gs and E values exhibited similar tendencies, except that the difference in gs between PEG- and salt-treated leaves was not significant.

The SPAD values of the leaves also decreased when the plants were exposed to osmotic stress, except in the case of PEG treatment. Chlorophyll degradation was observed, especially in plants treated with mannitol, followed by sorbitol and NaCl treatment. Chlorosis was not detected after PEG treatment (Table 1).

## Decrease in leaf water content and changes in the osmotic potential of leaf and root saps during osmotic stress

The exposure of young wheat plants to various osmotically active compounds reduced the leaf water content to different extents. RWC declined slightly after PEG or NaCl treatment but

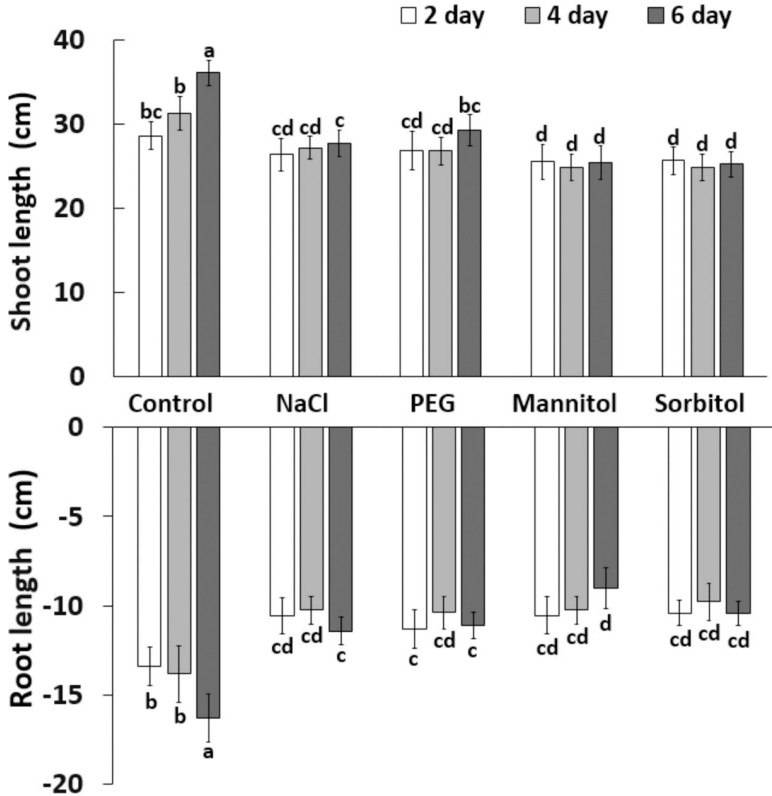

**Fig 1. Root and shoot length of plants (n = 20) grown in hydroponic solution containing NaCl, PEG, mannitol or sorbitol for 2, 4 or 6 days.** Data represent the mean ± STD. Different letters indicate significant differences at the $p < 0.05$ level using Tukey's *post hoc* test.

dropped substantially in leaves treated with mannitol or sorbitol (Fig 3). In these plants crystal formation was observed in guttated sap, especially after mannitol treatment (S1 Fig.).

The osmotic potential of both root and leaf sap decreased after treatment with NaCl, mannitol or sorbitol but the changes were less pronounced in the roots than in the leaves (Fig 4). In contrast, PEG treatment caused a decrease in the osmotic potential in the roots but it was unaffected in the leaves. In the roots, the most negative osmotic potential was detected after 6 days of NaCl treatment, followed by PEG treatment, whereas mannitol and sorbitol treatment only slightly modified the osmotic potential of root sap compared to that of the medium (-0.8 MPa). In leaves, however, mannitol and sorbitol caused a substantial, time-dependent decrease in the osmotic potential, similarly as it was found in NaCl-treated leaves and roots. These results suggest that different osmoregulatory processes may operate in the leaves and roots and that these processes also depend on the osmotically active agent applied. To determine what metabolites take part in osmoregulation, the amounts of several compatible solutes were measured from leaf and root sap.

## Metabolism during the osmotic stress induced by various compounds

Substantial differences were found in the concentration of proline, GB and sugar metabolites in tissue sap after various osmolyte treatment. The most pronounced proline accumulation was found in the roots and leaves of NaCl-treated plants (Fig 5A and 5B). Intense proline accumulation was also observed in the leaves of mannitol- and sorbitol-treated plants while PEG-treatment only caused a slight increase in proline in the leaves. In the roots, however, PEG,

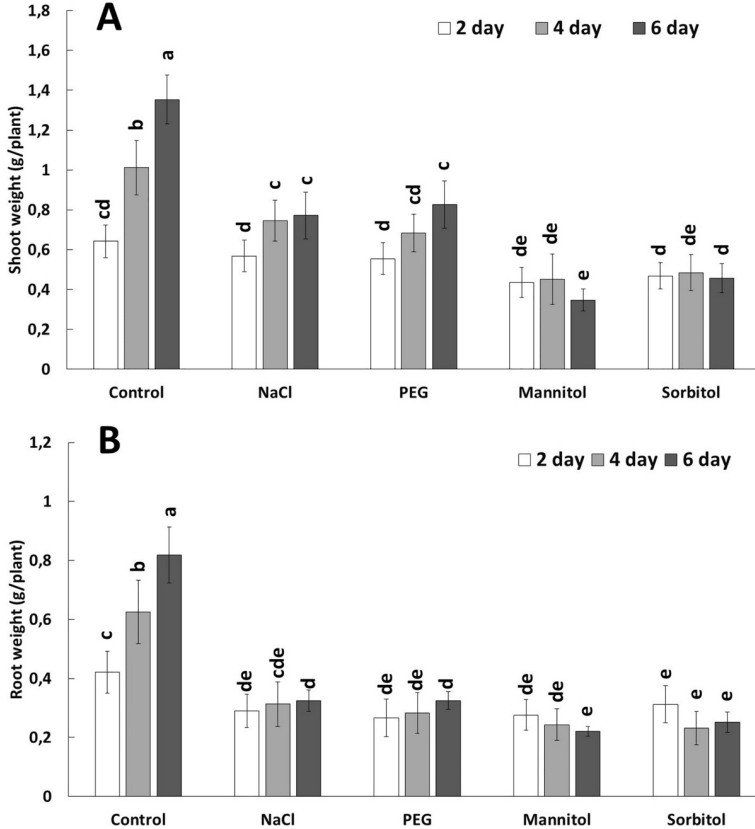

**Fig 2. Root and shoot weight of plants (n = 20) grown in hydroponic solution containing NaCl, PEG, mannitol or sorbitol for 2, 4 or 6 days.** Data show the mean ± STD. Different letters indicate significant differences at the p < 0.05 level using Tukey's *post hoc* test.

mannitol and sorbitol treatments all resulted in moderate proline accumulation. The glycine-betaine (GB) content in the roots was not influenced substantially by any of the treatments compared to the control, except that PEG caused extremely high GB accumulation (Fig 5C and 5D). In the leaves, a slight increase in GB was found in plants treated with NaCl, mannitol or sorbitol, but these changes were only temporary, as the GB content gradually decreased after 2 days of treatment.

With respect to total soluble sugars, the highest sugar accumulation was observed in root and leaf sap isolated from plants treated with mannitol or sorbitol (Fig 6). Soluble sugars

**Table 1. Photosynthetic activity (Pn), stomatal conductance (gs) and transpiration rate (E) in the leaves of plants grown in hydroponic solution containing NaCl, PEG, mannitol or sorbitol for 6 days (n = 5 for each treatment).** The SPAD values indicate the chlorophyll content of the leaves (n = 15 for each treatment).

| | Pn ($\mu$mol $CO_2$ m$^{-2}$ s$^{-1}$) | gs (mmol $H_2O$ m$^{-2}$ s$^{-1}$) | E (mmol $H_2O$ m$^{-2}$ s$^{-1}$) | SPAD value |
|---|---|---|---|---|
| **Control** | 19.6 ± 1.06 a | 282 ± 32 a | 3.4 ± 0.38 a | 39.0 ± 2.4 a |
| **NaCl** | 9.8 ± 0.88 c | 84 ± 17 b | 1.3 ± 0.34 c | 32.5 ± 2.0 b |
| **PEG** | 15.4±0.82 b | 121 ±24 b | 2.1±0.32 b | 39.8 ± 1.8 a |
| **Mannitol** | 7.31 ± 0.58 d | 25 ± 6.7 c | 0.6 ± 0.25 d | 28.1 ± 1.8 c |
| **Sorbitol** | 7.4 ± 0.72 d | 23 ± 5.8 c | 0.6 ± 0.17 d | 29.6 ± 1.6 bc |

Data show the mean ± STD; different letters indicate significant differences at the p < 0.05 level using Tukey's *post hoc* test.

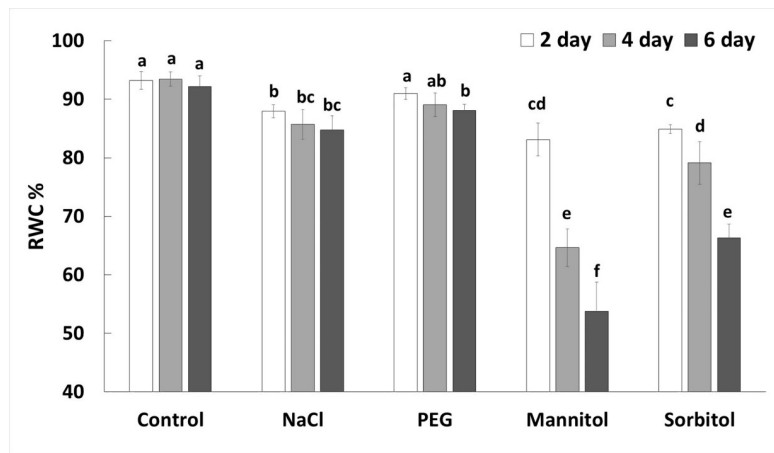

**Fig 3. Relative water content of leaves 2, 4 or 6 days after treatment with NaCl, PEG, mannitol or sorbitol.** Data are the means ± STD of 5 replicates for each day and treatment. Different letters indicate significant differences at the p < 0.05 level using Tukey's *post hoc* test.

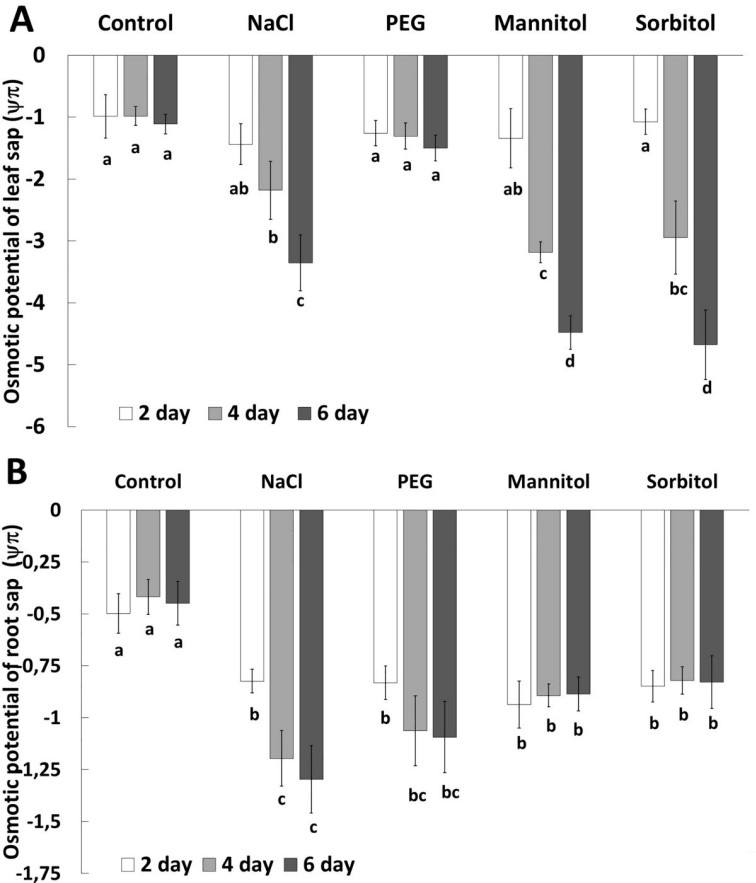

**Fig 4. Changes in the osmotic potential of leaf (A) and root (B) sap 2, 4 or 6 days after treatment with NaCl, PEG, mannitol or sorbitol.** Data are the means ± STD (n = 5 for each day and treatment); different letters indicate significant differences at the p < 0.05 level using Tukey's *post hoc* test.

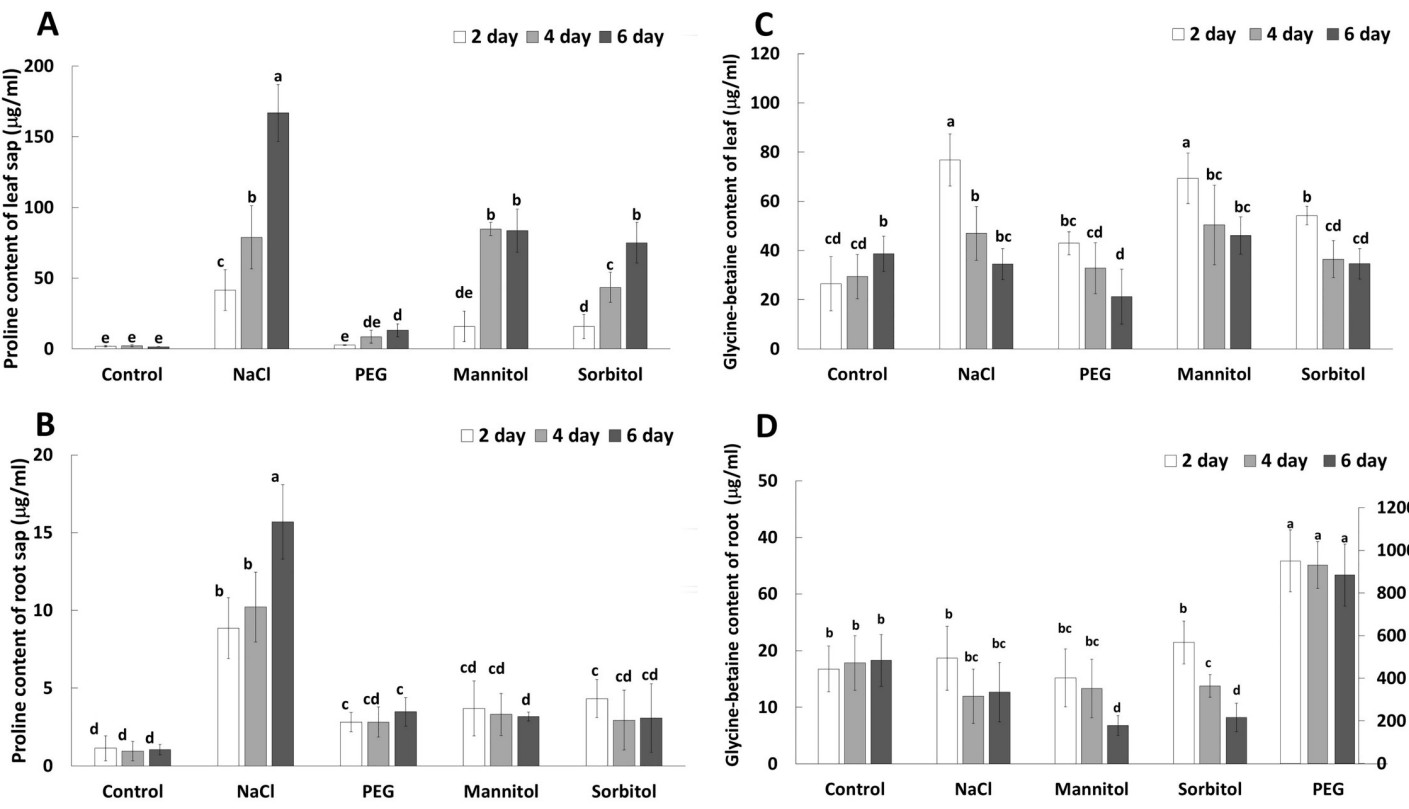

**Fig 5. Changes in the amount of proline (A,B) and QAC including GB (C,D) of tissue sap prepared from the leaves (A,C) and roots (B,D) of wheat plants grown in hydroponic solution containing NaCl, PEG, mannitol or sorbitol for 2, 4 or 6 days.** On figure D, the scale on the right refers to the PEG treatment. Data are the means ± STD of 5 replicates for each day and treatment and the different letters indicate significant differences at the $p < 0.05$ level using Tukey's *post hoc* test.

accumulated more slowly in the roots of plants treated with PEG or NaCl, not reaching the level found for mannitol and sorbitol on the 2[nd] day of treatment until the 6[th] day. In the leaves, however, lower amounts of sugar were found in PEG- and NaCl-treated plants than after mannitol or sorbitol treatments throughout the treatment period. The sugar composition also differed among the treatments. Salt treatment especially enhanced sucrose accumulation together with the induction of galactose synthesis in both roots and leaves. PEG treatment induced glucose, fructose and sucrose accumulation, especially in the roots. The main sugar compounds in leaf and root sap were mannitol and sorbitol when the plants were treated with these osmolytes, but fructose, sucrose and maltose could also be detected in both roots and leaves on the 2[nd] day of treatment. The amounts of these sugars decreased continuously in the roots but increased in the leaves as the treatment progressed. All these results demonstrated that the accumulation of compatible osmolytes was strongly dependent on the type of treatment and showed tissue specific responses.

PCA analysis (Fig 7) revealed the metabolic changes induced by various osmolytes separately for the root and shoot. The PCA of leaf yielded the most important components (PC1 and PC2) representing 59.9% of the total variance (Fig 7A). The score plots (Fig 7B) presented three clusters. Cluster I included both control and PEG-treated leaves due to lower negative values of osmotic potential and the accumulation of fructose and glucose. Cluster II concerned to mannitol- and sorbitol-treated leaves related to greater accumulation of maltose. Cluster III separated NaCl-treated leaves based on great amount of proline, sucrose and galactose. For

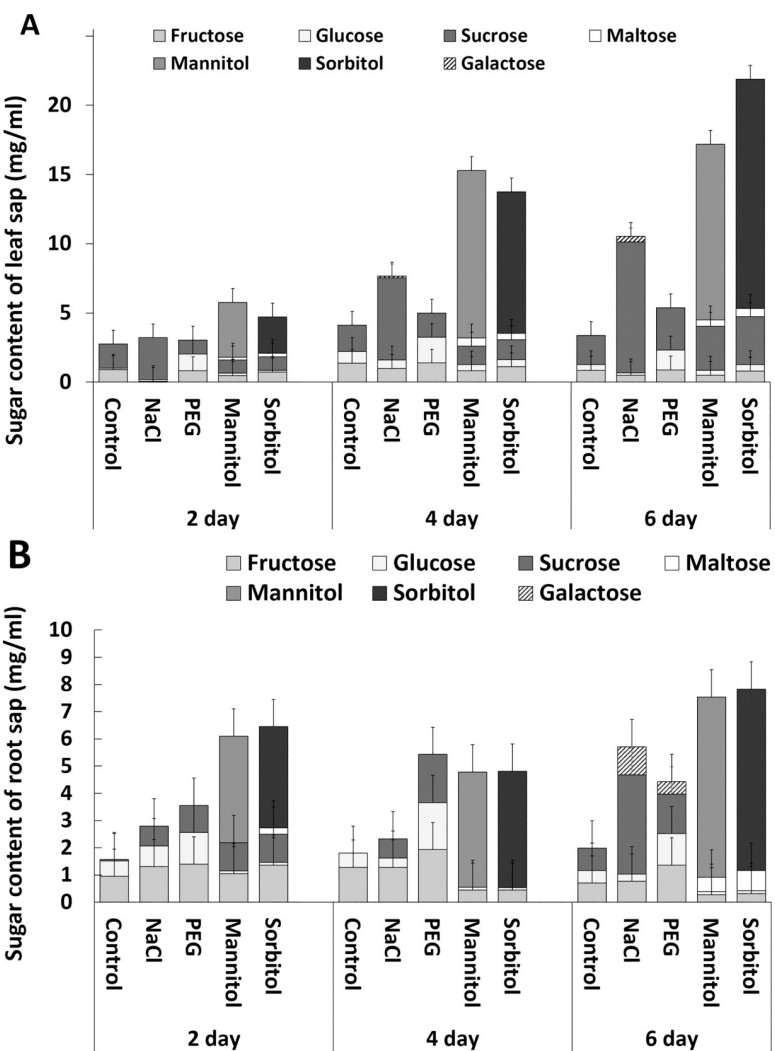

**Fig 6. Changes in sugar metabolites in the leaf (A) and root (B) sap of wheat plants grown in hydroponic solution containing NaCl, PEG, mannitol or sorbitol for 2, 4 or 6 days**. Data are the means ± STD (n = 5 for each day and treatment). The results of statistical analysis using Tukey's *post hoc* test are presented in S1 Table.

root, the total variation of two most important principal components (PC1 and PC2) was 63.68% of data variation (Fig 7C). The score plots demonstrated four distinct groups in root (Fig 7D). Group I included control while PEG-exposed roots separated in group II because of accumulation of GB, glucose and fructose. Cluster III belonged to mannitol and sorbitol stress, which had a similar influence on maltose synthesis while the cluster IV discriminated NaCl roots as separate group due to galactose, proline and sucrose accumulation. Factor loadings and variance analysis of various parameters for root and shoot are presented in S3–S6 Tables.

The polyamine metabolism was studied by determining free PAs (PUT, SPD and SPN) after exposing the plants to various osmolytes (Fig 8). The total amount of free PAs, especially the amounts of PUT and SPD increased in the leaves of mannitol- and sorbitol-treated plants and in the roots of PEG-treated plants. However, the relative amounts of PUT, SPD and SPN also changed in all the treated plants depending on the type of treatment. In NaCl-treated plants, the amounts of PUT and SPD decreased while that of SPN increased in both roots and leaves. After mannitol or sorbitol treatment, PUT and SPD increased considerably and SPN decreased in the leaves. PUT also increased in the roots but SPD and SPN decreased. PEG

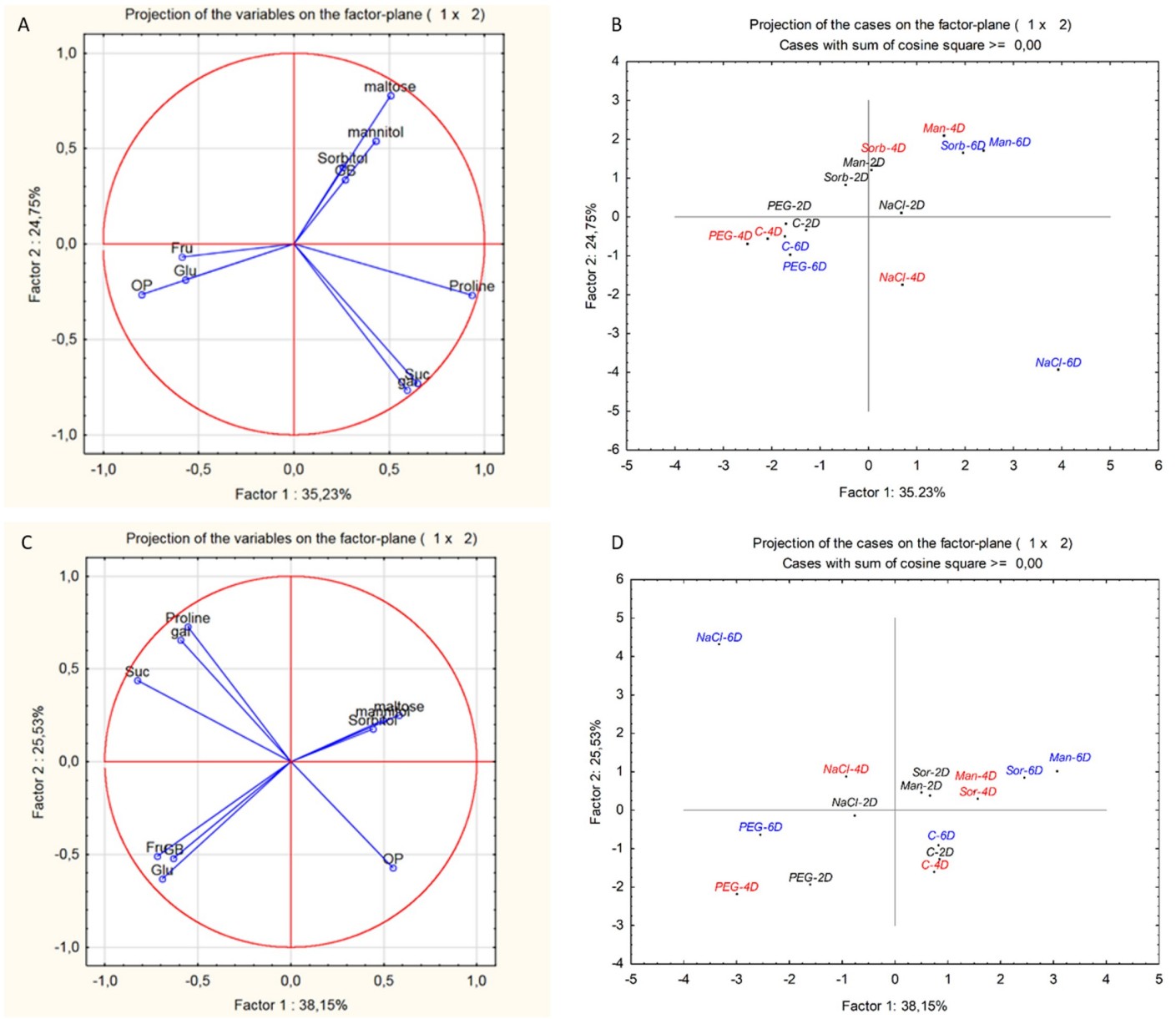

**Fig 7. Principal component analysis of root (A) and shoot (B) data sets. Loadings of the first two factors (C and D).** Principle component analysis (PCA) was applied for the evaluation of proline (Pro), glycine betaine (GB), osmotic potential (OP) and sugar metabolites (glucose-Glu; fructose-Fru; sucrose-Suc; galactose-Gal; and maltose-Malt) under control and different iso-osmotic stresses for 2, 4 and 6 days. Data were analysed by using STATISTICA 13.4 software package.

treatment did not modify the amounts of any of the PAs in the leaves as compared to the control but induced the accumulation of PUT, SPD and SPN in the roots. These results indicated that different osmotic treatments induced different processes in polyamine metabolism even under iso-osmotic stress conditions.

## Discussion

### Plant responses to various osmolytes

The responses of plants to osmotic stress induced either by PEG, mannitol, sorbitol or NaCl have been widely investigated separately, but comparative studies under iso-osmotic

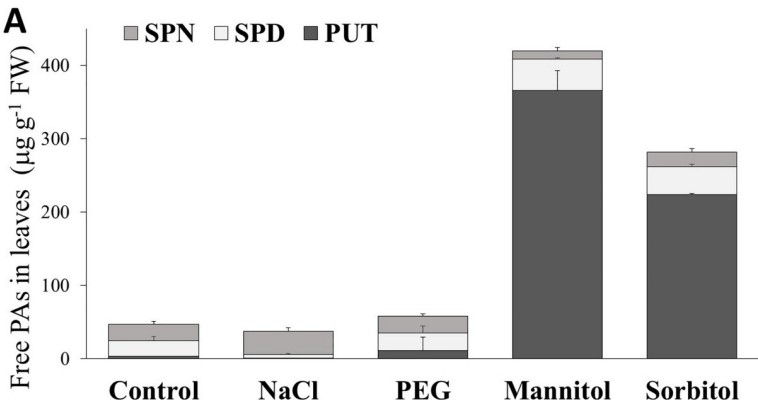

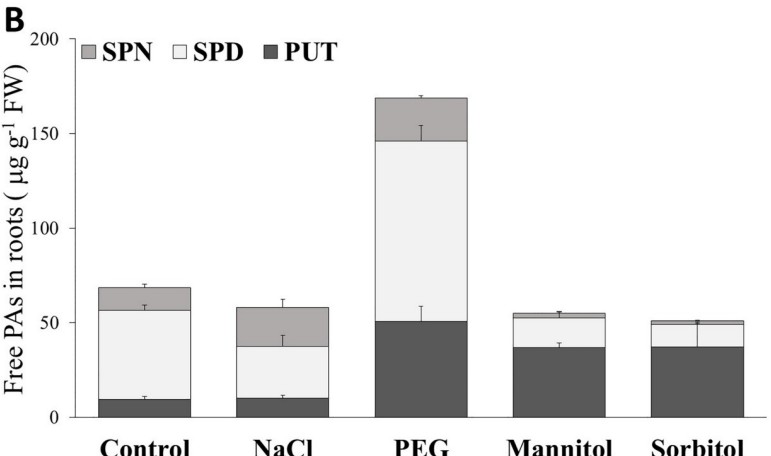

**Fig 8. Amounts of the free polyamines (PAs) putrescine (PUT), spermidine (SPD) and spermine (SPN) in the leaves and roots of wheat plants grown in hydroponic solution containing NaCl, PEG, mannitol or sorbitol for 6 days.** Data are the mean ± STD of 3 repetitions. The results of statistical analysis are presented in S2 Table.

conditions are rare [8,12]. The present results showed that although all the osmotic stress treatments inhibited growth immediately, as manifested in a reduction in both the length and weight of roots and shoots (Figs 1 and 2), the metabolic response of root and leaves depended on the nature of the osmotica.

A decrease in the osmotic potential of the culture media resulted in a reduction of the osmotic potential in the root sap too, but only NaCl treatment caused an additional decrease in osmotic potential over time (the osmotic potential values were lower than -0.8 MPa) (Fig 3). In the leaves, however, the osmotic potential decreased gradually in plants treated with NaCl, mannitol or sorbitol, showing that osmolyte accumulation increased over time. As indicated previously [12] NaCl, mannitol and sorbitol are able to penetrate into root cells and move towards the shoot and leaves (as also indicated by the guttation of mannitol through leaf pores; S1 Fig). The penetration of both non-ionic organic solutes (such as mannitol and sorbitol) and inorganic ions (e.g. NaCl) decreased the osmotic potential (Fig 3) in both the roots and leaves The same was demonstrated by Chen and Jiang [20]. In contrast, PEG molecules with higher than 3000 molecular weight cannot pass through the apoplastic barrier of root cells, and has no toxic effect within the cells while it reduces the water potential of the culture medium [21]. Accordingly, the PEG-induced decrease in osmotic potential was lower than that of the other osmolytes and was less pronounced in the leaves than in the roots. It is also related to the small

decreases in RWC, photosynthesis activity and chlorophyll content in the PEG-treated leaves. The metabolic responses induced by PEG also differed from those of the other osmolytes. PCA analysis showed separation of PEG-treated plants from mannitol-, sorbitol-or NaCl-treated plants. It is manifested in the accumulation of glucose and fructose both in the leaves and roots and that of GB in the roots. NaCl treatment mainly caused changes in proline, sucrose and galactose accumulation in both roots and leaves. Treatments with mannitol and sorbitol not only resulted in the high accumulation of the relevant osmolytes in both leaves and roots but also induced maltose accumulation, especially in the leaves.

### Role of GB and proline in osmoregulation

The role of GB, an important quaternary ammonium compound, in osmotic stress protection has been studied in detail. GB acts as an osmoprotectant in osmoregulation, it is involved in the stabilization of macromolecules, the maintenance of photosynthetic activity in leaves, and the scavenging of ROS [22]. In the present experiments, the amount of GB did not increase either in the leaves or in the roots after salt, mannitol or sorbitol treatment, so it cannot play an important role in osmotic adjustment, except in the case of PEG treatment after which a large amount of GB was accumulated in the roots. PCA also confirmed this observation. Since the osmotic potential inside of the cells is lower than in the solution, water can be absorbed from the nutrient solution. A similar conclusion was presented by Ghuge et al. [12].

Proline is an indispensable compound in studies related to osmotic stress. However, highlighting the role of proline in plant adaptation to osmotic stress is not always reliable. For instance, the proline content was similar when 150mM NaCl or 20% PEG were applied for 15 days in sugarcane [10] but it was higher in the leaves of a halophyte plant, *Sesuvium portulacastrum*, after 140 or 500mM mannitol treatment than after 10 or 20% PEG treatment [8]. In potato leaves, greater proline accumulation was observed in plants exposed to 100mM NaCl than in those exposed to 12% PEG or 100mM mannitol [12]. The present results are in accordance with the above [12] in spite of stronger stress (150mM NaCl, 20% PEG and 300mM mannitol and sorbitol) was applied in the present work. The highest proline accumulation was detected in the leaves of NaCl-treated plants, followed by mannitol and sorbitol and finally by PEG. Furthermore, a similar tendency was observed in the roots, except that the proline content of the roots in the PEG-treated plants was comparable to that of the roots of plants s treated with mannitol or sorbitol. According to Meloni et al. [23], proline cannot contribute significantly to the reduction in the osmotic potential, so it cannot play an important role in osmotic adjustment as an osmolyte. Instead, proline plays a role in the stabilization of cellular structures by forming a hydration shell around the proteins and/or by stabilization of redox potential [24]. The difference in the proline accumulation between the salt-treated plants and others also support the view that proline is more inducible by ionic than by osmotic effect of salt.

### Role of sugars in osmoregulation

It is widely documented that osmotic stress triggers the accumulation of soluble sugars, such as glucose, fructose, sucrose, galactose and trehalose [3]. These sugars take part in osmoregulation and in the stabilization of protein structure enabling them to maintain their functions. They serve as metabolic precursors for other metabolites or together with other metabolites, such as GABA or polyamines, may act as signalling molecules, controlling the gene expression in various metabolic pathways [25]. Yasseen et al. [26] concluded that sugars can contribute to about 30–50% of the osmotic adjustment in the leaves of many glycophytes. In the present work, we found that sugar accumulation also depended on the nature of the osmotica. PEG

treatment mainly influenced the fructose and glucose metabolism in both roots and leaves. NaCl treatment was linked to the accumulation of sucrose and galactose while, not surprisingly, the uptake and transport of mannitol and sorbitol were the dominant factors in mannitol- and sorbitol-treated plants, respectively.

Glucose and sucrose are the main products of the photosynthetic assimilates in leaves. Under stress conditions, $CO_2$ assimilation generally decreases while respiration increases resulting in the accumulation of glucose and fructose at the expense of sucrose. However, in the present experiments the increase in glucose and fructose was accompanied by an increase in sucrose during PEG treatment. In addition, the decrease in $CO_2$ assimilation was less intense in the PEG-treated plants than in the case of other osmolytes. These results suggest that the higher levels of glucose and fructose were not caused by increased in catabolic processes in PEG-treated plants.

In the case of salt treatment, sucrose accumulation was the most pronounced change in the sugar metabolism. Similarly, when Gill et al. [27] compared the sugar composition of sorghum exposed to salt or PEG treatment, higher sucrose content was detected in salt-treated plants while the glucose and fructose content was higher in PEG-treated plants. The present results are in agreement with Gavaghan et al. [6], who reported that salt treatment inhibited the conversion of sucrose to glucose and fructose in both the roots and leaves of maize seedlings.

It is worth mentioning that salt stress also induced galactose accumulation as previously observed in other barley and wheat plants [28]. It is documented that L-galactose plays an important role in the ascorbic acid pathway and may contribute to the elevated salt tolerance of transgenic rice genotypes [29]. The present results also confirmed the importance of galactose in the salt stress response.

As NaCl causes both ionic and osmotic stress, salt treatment is thought to have a more drastic effect on plants than PEG, which cannot penetrate the cells, or mannitol and sorbitol, which penetrate the cells but are non-ionic. Under iso-osmotic stress conditions, this was evident when the PEG and NaCl treatments were compared (as indicated by the greater chlorophyll degradation and the decrease in photosynthetic parameters and RWC), but the mannitol and sorbitol treatments resulted in a more intense decrease of chlorophylls and RWC than the other treatments. This was mainly due to the high accumulation of mannitol and sorbitol in both leaves and roots. The concentrations of these metabolites were so high that the contribution of other metabolites to the decrease in osmotic potential seems marginal. On the other hand, the PCA revealed that mannitol and sorbitol accumulation was related to the amount of maltose. An elevated maltose content was also detected in Arabidopsis plants exposed to osmotic stress induced by mannitol treatment due to the elevated activity of β- and α-amylase, which is regulated by the thioredoxin and ABA pathways [30,31].

## Changes in the polyamine metabolism under iso-osmotic stress conditions induced by various osmolytes

Many papers have demonstrated that polyamines are involved in plant stress responses and tolerance (see review in Pál et al. [32]). The amounts of the main polyamines, PUT, SPD and SPN, are strongly dependent on their synthesis and catabolism. In addition, the PA metabolism is interconnected with other stress-related metabolic routes including proline metabolism and also influenced by various plant hormones and growth regulators [33]. In spite of the huge number of investigations, the role of PAs in plant stress processes are often contradictory [32].

In the present experiments, the amounts of PAs, especially PUT and SPD, increased significantly in the leaves of mannitol- or sorbitol-treated plants and in the roots of PEG-treated plants. Previous studies indicated that the ornithine, arginine and S-adenosyl-methionine

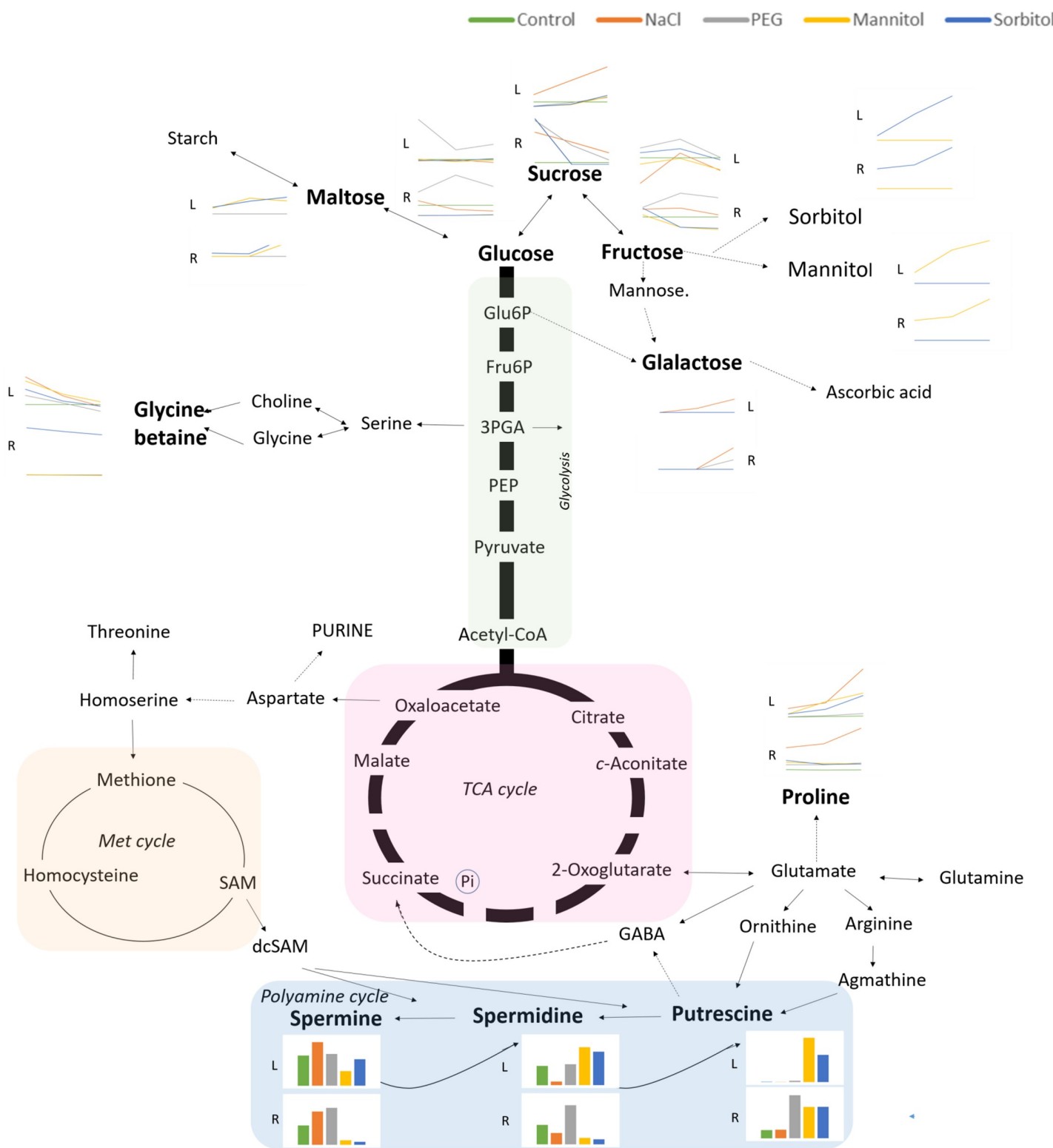

**Fig 9. Metabolic pathway indicating the metabolites associated with the metabolism of various identified sugars, amino acids and polyamines in wheat during iso-osmotic stress induced by various osmolytes.** Metabolites in black was detected. The dominant effects induced by PEG (grey), NaCl (orange) mannitol (yellow) or sorbitol (blue) are indicated above the metabolites. Green color shows the control values.

(SAM) metabolic routes were found to be active in drought-treated plants [34]. Since the osmotic adjustment induced by mannitol and sorbitol is not forced by changes in the sugar metabolism or by the induction of proline accumulation, it is possible that these osmolytes (mannitol and sorbitol) stimulated the PA pathways by activating the synthesis of PUT from precursors such as ornithine, arginine and the amount of SPD from SAM [34]. However, further investigations are required to prove the activation of these metabolic routes after mannitol, sorbitol or PEG treatment.

During salt stress, the size of the PA pool (the amount of PUT, SPD and SPM) did not change but the PAs underwent rapid interconversion. The proportion of PUT and SPD decreased while the amount of SPN increased in both roots and leaves. SPN accumulation was also observed in the leaves of rice, barley and other wheat cultivars after salt treatment [4,28]. The present results show that the activation of PA metabolic pathways depends not only on the severity of stress and sensitivity of plants but also on the nature of osmolytes.

## Conclusions

A comparative study on the metabolic changes under controlled conditions, independently of environmental factors, could contribute to a better understanding of the plant responses to osmotic stress. Fig 9 shows the metabolic pathways associated with the identified metabolites together with the dominant osmolytes affecting them (Fig 9).

PEG treatment resulted in the accumulation of glucose and fructose in both roots and leaves and that of GB in the root. Mannitol and sorbitol treatments mainly decreased the osmotic potential through the uptake, transport and accumulation of mannitol and sorbitol treatments, respectively and to a certain extent through that of maltose leading to most intense decrease in the RWC, chlorophyll and photosynthetic activity of the leaves. Salt treatment mainly caused the accumulation of proline, sucrose and galactose. The various osmolytes affected the polyamine metabolisms in different ways: mannitol and sorbitol treatments activated the polyamine metabolism in the leaves resulting in an elevated amount of PUT and SPD while all PAs increased in the roots of PEG-treated plants. The amount of PUT and SPD decreased while SPN increased after salt treatment both in the roots and leaves. These results show that the various osmolytes activated different metabolic processes even under iso-osmotic stress conditions and these changes also differed in the leaves and roots.

## Supporting information

**S1 Fig. Crystal formation in leaves when the plants were exposed to mannitol treatment.** (JPG)

**S1 Table. Results of statistical analysis for the changes of sugar metabolites presented in Fig 6.** Different letters indicate significant differences at p < 0.05 level using Tukey's *post hoc* test. The results are based on five biological replicates for each treatment and day. (DOCX)

**S2 Table. Results of statistical analysis for free polyamines, putrescine (PUT), spermidine (SPD) and spermine (SPN) determined from the leaves and root, as presented in Fig 8.** Different letters indicate significant differences at p < 0.05 level using Tukey's *post hoc* test. The results are based on three biological replicates for each treatment and day. (DOCX)

**S3 Table. Analysis variance of various physiological parameters for shoot.** Data were analysed by using STATISTICA 13.4 software package. (DOCX)

**S4 Table. Analysis variance of various physiological parameters for root.** Data were analysed by using STATISTICA 13.4 software package.
(DOCX)

**S5 Table. Factor loadings of sugars, proline, GB and osmotic potential.** Principle component analysis (PCA) was applied for evaluation of response of shoots under control and osmotic stresses. Data were analysed by using STATISTICA 13.4 software package.
(DOCX)

**S6 Table. Factor loadings of sugars, proline, GB and osmotic potential.** Principle component analysis (PCA) was applied for evaluation of response of roots under control and osmotic stresses. Data were analysed by using STATISTICA 13.4 software package.
(DOCX)

## Acknowledgments

This work was funded by the National Research, Development and Innovation Office, grant No. K112226.

## Author Contributions

**Conceptualization:** Eva Darko.

**Data curation:** Tihana Marček.

**Investigation:** Eva Darko, Balázs Végh, Radwan Khalil, Gabriella Szalai, Magda Pál.

**Visualization:** Balázs Végh, Tihana Marček.

**Writing – original draft:** Eva Darko, Tihana Marček, Tibor Janda.

**Writing – review & editing:** Eva Darko.

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
