## [Decision Letter · Decision Letter 0]

28 Oct 2019

PONE-D-19-24786

Metabolic responses of wheat seedlings to osmotic stress induced by various osmolytes under iso-osmotic conditions

PLOS ONE

Dear Dr. Eva Darko,

Thank you for submitting your manuscript to PLOS ONE. After careful consideration, we feel that it has merit but does not fully meet PLOS ONE’s publication criteria as it currently stands. Therefore, we invite you to submit a revised version of the manuscript that addresses the points raised during the review process.

We would appreciate receiving your revised manuscript by 09/11/2019. To enhance the reproducibility of your results, we recommend that if applicable you deposit your laboratory protocols in protocols.io, where a protocol can be assigned its own identifier (DOI) such that it can be cited independently in the future. For instructions see: http://journals.plos.org/plosone/s/submission-guidelines#loc-laboratory-protocols

We look forward to receiving your revised manuscript.

Kind regards,

Mayank Gururani

Academic Editor

PLOS ONE

Journal Requirements:

Reviewers' comments:

Reviewer's Responses to Questions

**Comments to the Author**

1. Is the manuscript technically sound, and do the data support the conclusions?

Reviewer #1: Yes

Reviewer #2: Partly

2. Has the statistical analysis been performed appropriately and rigorously? 

Reviewer #1: Yes

Reviewer #2: Yes

3. Have the authors made all data underlying the findings in their manuscript fully available?

Reviewer #1: Yes

Reviewer #2: No

4. Is the manuscript presented in an intelligible fashion and written in standard English?

Reviewer #1: No

Reviewer #2: No

5. Review Comments to the Author

Reviewer #1: Dear Author

The work submitted to PLOSone requires the following

1. please provide the the number of sample analysed for each trait in the manuscript as n=...., because it is so confusing. HTe experiment was repeated thrice and it has 5-10 biological samples and after that there are 5 biological replicates. Also indicate exact number in some places, 5 or 15 samples were analysed in some places 5 or 10..

2. The method for GB (spectrophotometer) has its own limitation becasue lugol can also forms colour with other metabolite, and hence it can be removed

3. please provide information on design of experiment and how the treatments were arranged. The graphs presented are interaction graphs (treatment and time), then please provide the anova to show whether there is any day effect.

4. Please explain the statistical procedure followed in PCA, also the need for cosine transformation, and rotation of axis.

5. Figure quality may be improved.

6. Figure 9 needs to be modified by showing the % increase instead of trend line.

7. Procedure of PA analysis may be provided in detail, along with efficiency of extraction. Whether internal standards are included during extraction, authenticity of standard (purity).

8. Discussion may be reduced significantly by removing the speculations

Reviewer #2: The following points should be addressed before accepting the paper

1. The aim of the experiment is not very convincing that why such type of study is required and what the benefit the readers have and how this will benefited to the agriculture community

2. The quality of the figures are very bad and hazy. It should be revised

3. It has 57 reference and some of them are not relevant. Try to reduce the reference to a level of 40. Cite the new reference.

4. Abstract should be rewritten. A concluding remark that how this study will be helpful is to be added.

5. The experiment is under stress therefore the antioxidative enzymes such as peroxidase, catalase, superoxide dismutase etc should be done to strengthen the data and conclusion

In my opinion it can not be accepted in the present form

6. PLOS authors have the option to publish the peer review history of their article (what does this mean?). If published, this will include your full peer review and any attached files.

Reviewer #1: No

Reviewer #2: Yes: SHAMSUL HAYAT

---

## [Author Response · Author response to Decision Letter 0]

8 Nov 2019

Dear Editor and Reviewers, 

First of all, we would like to thank you and the referees for thoroughly revision of the manuscript and for their critical comments. We strived to correct all mistakes and we hope that according to the instruction of reviewers, the manuscript has improved considerably. Thank you very much. 

Answers to reviewers: 

Reviewer #1: Dear Author

The work submitted to PLOSone requires the following

1. please provide the the number of sample analysed for each trait in the manuscript as n=...., because it is so confusing. HTe experiment was repeated thrice and it has 5-10 biological samples and after that there are 5 biological replicates. Also indicate exact number in some places, 5 or 15 samples were analysed in some places 5 or 10..

Thank you very much for the comments The Materials and method section is modified according to the request. The number of repetitions was indicated separately for each measurement. In addition, the figure legends were also amended. We hope that the number of repetitions will not be confusing in this way. Thank you very much. 

2. The method for GB (spectrophotometer) has its own limitation becasue lugol can also forms colour with other metabolite, and hence it can be removed

Yes, the reviewer is right that lugol can give reaction with several molecules. Starch forms blue colour with lugol but in our case, the supernatants of tissue saps were used for the investigations, which containts only small and water soluble metabolites. It is also true that the KI-I2 reagent produces crystals with several quaternary ammonium compounds, including the major compound GB. The manuscript has been corrected according to this. As in many papers (e.g. Valadez-Bustos et al. (2016) Analitical Biochemistry, 498, 47-; Habib et al. (2012, South African Journal of Botany 83, 151-), we also used this method (method of Grieve and Grattan (Rapid assay for determination of water soluble quaternary ammonium compounds. Plant Soil. 1983;70: 303–307. doi:10.1007/BF02374789) for the determination of GB from plant tissues because it is simpler than other chromatographic methods. However, in the future, we will use other method, e.g. chromatographic method, which separates the quaternary ammonium compounds. 

3. please provide information on design of experiment and how the treatments were arranged. The graphs presented are interaction graphs (treatment and time), then please provide the anova to show whether there is any day effect. 

As requested by the reviewer, we prepared a figure explaining our experimental design (please find it in the attachment). In addition, we also performed the ANOVA to make sure that the results and our conclusions are valid. We present these information as supplementary data. 

4. Please explain the statistical procedure followed in PCA, also the need for cosine transformation, and rotation of axis.

After normalization of data, PCA was performed. Measurement variables fit a normal distribution. PCA analysis was based on correlation matrix. The data set used for PCA consisted of 10 variables. PCA was applied to the standardized data set and the factor loadings were done in order to estimate the proportion of total variance with different principal components. The loadings showed correlations among different principal components (PC) and measurable variables whereby high loadings represented strong correlation (>0.75). Dropping factors whose eigenvalues were less than 1, were not included in analysis. Moreover, we used the criterion for PCA usage if the total variance of the first two factors was higher than 50%. 

Factor loadings with variable contribution and eigenvalues was presented in Supplementary Tables 5 and 6. 

For shoot, the PCA yielded the total variation of four principal components showing 91.2% of data variation under control and osmotic stress treatments (STable 5). The most important were two components explaining 58.76% of data variance. The first component (PC1) was largely determined by high negative loadings on osmotic potential while proline and sucrose had high positive loadings. The second component (PC2) was largely determined by high positive loadings related to maltose and sorbitol, respectively. GB had high positive loadings in the third component (PC3) and fourth component (PC4) reveals negative loadings on mannitol. 

For root, the PCA yielded the total variation of three principal components showing 76.8% of data variation under control and osmotic stress treatments (STable 6).The total variation of the two most important principal components (PC1 and PC2) was 63.68% of data variation. PC1 was determined by high positive loadings on osmotic potential and negative loadings on proline, sucrose, galactose, maltose, mannose and sorbitol, respectively. Fructose, glucose and GB had positive loadings in PC2 while positive loadings of sorbitol and negative loadings of mannitol contributed in PC3. 

With cosine transformation, we can orthogonalize a certain vector and decrease its dimensionality. PCA transformation is useful because the first few signal components highly correlate and contain the same content of the data energy. Rotation of axis makes components decorrelated and information is concentrated in small number of components. 

5. Figure quality may be improved.

We also noticed that the quality of the figures changed during the PDF conversion, in spite of the fact that the figures have been prepared in TIFF at 600 dpi. We reconstructed the figures and they are re-uploaded. If they are still in bad quality, please download the new files. 

6. Figure 9 needs to be modified by showing the % increase instead of trend line. 

As requested by the reviewer, the data are recalculated and a new figure 9 was prepared showing the % increase instead of trend line. (Please find the new figure attached.) In these cases, the control values are always 1 and show linear line (green) in the figures. In most of the cases, the tendencies remained the same however, this type of presentation resulted in discrepancies in some cases. For example, in the roots, the sucrose content increased slightly during the time in the control plants (from 0.07 to 0.6), which is reasonable. In salt–treated plants the sucrose content is always higher than those of control and shows an increasing tendency as sucrose content changed from 0.7 to3.7 during the time. However, when we calculated the changes in % the results showed opposite tendencies (the salt-induced change was decreased in time). It was only due to the mathematic calculation (as for example 1.0/0.1 =10; while 3.7/0.6 = 6 ), but it is not a biological reason. The other problem occurred during the calculation, when the amount of a special compound was under or close to the detectable limit in control plants. For example the amount of mannitol and sorbitol remained below the detectable threshold in the healthy (control) plants but, of course, their amount increased significantly when the plants were treated with them, and these metabolites transported to the root and shoot. In those cases when the level of metabolites were close to the threshold, we gave 0 values for these metabolites. But mathematically, any number cannot be divided by zero. 

Since we discussed only those results, which presented significant changes and clear tendencies, in our opinion, it would be better to leave the original figure in the manuscript instead of a new one. Of course, if the reviewer insists, the old figure will be replaced with the new one. 

7. Procedure of PA analysis may be provided in detail, along with efficiency of extraction. Whether internal standards are included during extraction, authenticity of standard (purity).

In these experiments, we did not use internal standard for the determination of the amount of polyamines. Determination of polyamines by HPLC is a routine procedure in our laboratory therefore, we check regularly the injector precision and the detector accuracy/stability with the use of external standards purchased from Sigma-Aldrich (Darmstadt, Germany). External standards of PUT, SPD and SPN were also used for the quantification. However, the information was missed from the materials and methods. Thank you very much for drawing our attention to it. The manuscript is completed with this description. 

Several years ago, we checked the efficiency of extraction method by the use of internal standard, norvaline and we found our method appropriate. Same tendency was obtained by the use of internal or external standards. In the present experiments, the same isolation process was used during the whole experiments and it is likely that the extraction rate of each polyamine and the sensitivity of the detector and the precision of the injector did not change during this short period of investigation. Therefore, we suppose that our method using external quantification is suitable for the determination of polyamine contents of leaves and root. Of course, we agree that the utilization of an internal standard is useful (although it is not always applied). We will use it in the future again. 

8. Discussion may be reduced significantly by removing the speculations

We reduced the text by removing the speculation. 

Finally, we would like to thank you again for the evaluation of the manuscript, for valuable recommendations and suggestions. We were about to correct all the critical remarks and we hope that according to the instruction of reviewer, the manuscript has imrpoved considerably. Thank you very much. 

Reviewer #2: The following points should be addressed before accepting the paper

1. The aim of the experiment is not very convincing that why such type of study is required and what the benefit the readers have and how this will benefited to the agriculture community

First of all, we wish to thank you and the referees for the evaluation of the manuscript, We appreciate for the valuable recommendations and suggestions. 

Studying the abiotic stress processes affecting the yield production is really important for us. Plant responses to drought and salt stress have been widely investigated in the world and also at our department too. Many articles have been published on this topic, and we realized that the results strongly depend on conditions applied (e.g. on the type of treatment, plant sensitivity, and other environmental factors). Beasides, as we presented in the introduction, the results (not only ours but also others’) are often contradictory. The explanation is difficult due to the completely different circumstances (e.g. growth conditions, species or genotypes). 

In the present experiments, we aimed to identify the common and specific metabolic responses to osmotic stress induced by different osmolytes including salt or PEG, mannitol or sorbitol, which are often used to stimulate drought stress in plants. We kept as many conditions at the same level as possible meanwhile we were studying several physiological and metabolic parameters. Yes, the reviewer is right that this is a simplified (model) system in comparison to natural environmental conditions. However, our results identified several different metabolic responses (eg proline - NaCl; Peg - GB, or PAs). We believe, that these information can be useful for all scientists who study those processes which trigger osmotic stress. 

2. The quality of the figures are very bad and hazy. It should be revised

We also noticed that the quality of the figures changed during the PDF conversion, in spite of the fact that the figures had been prepared in TIFF at 600 dpi. We reconstructed the figures and they are re-uploaded. If they are still in bad quality, please download the new files. 

3. It has 57 reference and some of them are not relevant. Try to reduce the reference to a level of 40. Cite the new reference.

According to the comments of both reviewers, the text modified significantly and we also checked and reduced the number of references. Thank you very much for your suggestions. 

4. Abstract should be rewritten. A concluding remark that how this study will be helpful is to be added.

The abstract is rewritten and placed in a bigger context to emphasize its importance. 

5. The experiment is under stress therefore the antioxidative enzymes such as peroxidase, catalase, superoxide dismutase etc should be done to strengthen the data and conclusion

In general, many stresses including salt or drought induce osmotic stress in plants. These stresses cause oxidative damage (as secondary stress) in plants, especially if they are severe. In many cases, the oxidative stress is only a consequence of the primary stress. In many of our previous investigations, we also studied the antioxidant systems under different environmental conditions (salt, drought, cold, heat or metal toxicity), and we found that in some cases, such as cold or heavy metal toxicity, the antioxidant system plays an important role in the protection against the primary stress. However, in other cases, including salt or drought their role in the plant responses is less pronounced compared to the changes of other metabolites (Darko et al. 2004, Darko et al, 2009 Crop Sci, ; Darko et al. PlosOne, 2017, Janda et al. ). Therefore, in the present investigations we did not compare the changes of the antioxidant systems. Instead, we focused on following the metabolic changes under osmotic stress induced by salt or other compounds generally used to simulate drought stress. However, we eliminated those explanation from the manuscript which were related to the antioxidant systems, since as you are right, they would be only speculations without supporting investigations behind them. 

Finally we would like to thank you again your work. Your thorough revision and critical comments have been valuable for us. We were about to correct all the critical remarks and we hope that according to the instruction of reviewer, the manuscript has improved considerably. Thank you very much.

---

## [Decision Letter · Decision Letter 1]

21 Nov 2019

Metabolic responses of wheat seedlings to osmotic stress induced by various osmolytes under iso-osmotic conditions

PONE-D-19-24786R1

Dear Dr. Eva Darko,

We are pleased to inform you that your manuscript has been judged scientifically suitable for publication and will be formally accepted for publication once it complies with all outstanding technical requirements.

With kind regards,

Mayank Gururani

Academic Editor

PLOS ONE

Additional Editor Comments (optional):

Reviewers' comments:

Reviewer's Responses to Questions

**Comments to the Author**

1. If the authors have adequately addressed your comments raised in a previous round of review and you feel that this manuscript is now acceptable for publication, you may indicate that here to bypass the “Comments to the Author” section, enter your conflict of interest statement in the “Confidential to Editor” section, and submit your "Accept" recommendation.

Reviewer #1: All comments have been addressed

Reviewer #2: All comments have been addressed

2. Is the manuscript technically sound, and do the data support the conclusions?

Reviewer #1: Yes

Reviewer #2: Yes

3. Has the statistical analysis been performed appropriately and rigorously? 

Reviewer #1: Yes

Reviewer #2: Yes

4. Have the authors made all data underlying the findings in their manuscript fully available?

Reviewer #1: Yes

Reviewer #2: Yes

5. Is the manuscript presented in an intelligible fashion and written in standard English?

Reviewer #1: Yes

Reviewer #2: Yes

6. Review Comments to the Author

Reviewer #1: No comments - All the queries has been addressed

In future, highlight the changes in the revised manuscript for easy understanding.

Reviewer #2: Authors have thoroughly address all the comments. The suggestion has been incorporated as required. Therefore. I have no hesitation in recommending the paper for publication in this journal.

Accepted for publication

7. PLOS authors have the option to publish the peer review history of their article (what does this mean?). If published, this will include your full peer review and any attached files.

Reviewer #1: No

Reviewer #2: No

---

## [Editor Report · Acceptance letter]

6 Dec 2019

PONE-D-19-24786R1 

Metabolic responses of wheat seedlings to osmotic stress induced by various osmolytes under iso-osmotic conditions 

Dear Dr. Darko:

I am pleased to inform you that your manuscript has been deemed suitable for publication in PLOS ONE. Congratulations! Your manuscript is now with our production department. 

With kind regards,

on behalf of

Dr. Mayank Gururani 

Academic Editor

PLOS ONE